# AMFF-Net: An Effective 3D Object Detector Based on Attention and Multi-Scale Feature Fusion

**DOI:** 10.3390/s23239319

**Published:** 2023-11-22

**Authors:** Guangping Li, Zuanfang Mo, Bingo Wing-Kuen Ling

**Affiliations:** School of Information Engineering, Guangdong University of Technology, Guangzhou 510006, China; 2112103208@mail2.gdut.edu.cn (Z.M.); yongquanling@gdut.edu.cn (B.W.-K.L.)

**Keywords:** 3D object detection, LiDAR, point cloud, multi-scale feature fusion, attention mechanism

## Abstract

With the advent of autonomous vehicle applications, the importance of LiDAR point cloud 3D object detection cannot be overstated. Recent studies have demonstrated that methods for aggregating features from voxels can accurately and efficiently detect objects in large, complex 3D detection scenes. Nevertheless, most of these methods do not filter background points well and have inferior detection performance for small objects. To ameliorate this issue, this paper proposes an Attention-based and Multiscale Feature Fusion Network (AMFF-Net), which utilizes a Dual-Attention Voxel Feature Extractor (DA-VFE) and a Multi-scale Feature Fusion (MFF) Module to improve the precision and efficiency of 3D object detection. The DA-VFE considers pointwise and channelwise attention and integrates them into the Voxel Feature Extractor (VFE) to enhance key point cloud information in voxels and refine more-representative voxel features. The MFF Module consists of self-calibrated convolutions, a residual structure, and a coordinate attention mechanism, which acts as a 2D Backbone to expand the receptive domain and capture more contextual information, thus better capturing small object locations, enhancing the feature-extraction capability of the network and reducing the computational overhead. We performed evaluations of the proposed model on the nuScenes dataset with a large number of driving scenarios. The experimental results showed that the AMFF-Net achieved 62.8% in the mAP, which significantly boosted the performance of small object detection compared to the baseline network and significantly reduced the computational overhead, while the inference speed remained essentially the same. AMFF-Net also achieved advanced performance on the KITTI dataset.

## 1. Introduction

With the dramatic growth of computer vision applications, especially autonomous driving technology, the requirements of vehicle perception of the surrounding environment are increasing. Point cloud-based 3D-object-detection techniques have received much attention from both industry and academia. The main sensors commonly deployed in intelligent vehicles are cameras and LiDAR. However, the detection performance of the camera is often affected by the environment, such as in bright light, night, rain and fog, etc., for which the camera detection performance is extremely poor [1]. As increasingly affordable and advanced LiDAR technology becomes available, 3D object detection has been integrated as an inseparably part of self-driving cars [2]. Compared to traditional camera-based systems, LiDAR’s ability to circumvent ambient noise and interference is unparalleled, and its ability to capture precise and structured point cloud data makes it an invaluable tool for determining the location and geometric attributes of relative objects. It forms the basis of the complex decision-making and planning processes that underpin autonomous driving [3]. However, despite the many advantages of LiDAR-based 3D object detection, significant challenges must be overcome to achieve optimal accuracy. The inherent disorder, sparsity, and inhomogeneous distribution of the point cloud pose significant obstacles to accurate and reliable detection, requiring sophisticated algorithms and innovative approaches to boost object-detection performance [4].

Given the numerous obstacles associated with point cloud 3D object detection, a wide range of groundbreaking algorithms have emerged in recent years. There are two main categories that dominate the field of 3D object detection: point-based methods and voxel-based methods. Point-based methods directly take raw point data as the network input and extract the features of each individual point within the point cloud scenes. The unique advantage of this approach is the ability to utilize the inherent geometric structure information of the point cloud, thus facilitating highly accurate object detection. However, the bottleneck of this method is the inference time, which requires the sample points to be collected by the Farthest Point Sampling (FPS) [5]. It suffers from the disadvantages of time-consuming and slow detection, which makes it difficult to be applied to real-time detection scenarios. As the field of 3D object detection continues to evolve, voxel-based methods (e.g., VoxelNet [6] and PointPillars [7]) have become prominent, characterized by their ability to rasterize the point cloud into a discrete grid representation. This process involves dividing the point cloud into voxels, columns, and Bird’s-Eye View (BEV) feature maps, which are then rigorously analyzed using either 2D convolutional neural networks or 3D sparse neural networks. However, the limitation of voxel-based approaches is that converting point cloud data into voxels via encoding can result in a loss of critical feature information, which affects the accuracy of object detection, especially for small objects. HotSpotNet [8] transforms voxel features into a novel representation, reducing the complexity and irregularity of the input data. However, this conversion also leads to a loss of detailed information from the original point cloud, which can be detrimental to small object detection. In addition, the loss of fine-grained localization accuracy is an important issue as it impairs the ability of 3D-object-detection systems to work effectively in the real world. Compared to existing voxel-based methods, which solely utilize single-scale voxel representation, Voxel-FPN [9] encodes multi-scale voxel features. However, while this method improves fine-grained localization accuracy, it introduces additional computational overhead to the model. As the field of detection technology continues to advance at a breakneck pace, the question of whether voxel-based detection methods can achieve point-based detection performance while maintaining existing detection speed has emerged as a formidable challenge.

The voxel-based methods divide the point cloud spatially into voxels and subsequently transform the point cloud features into voxel features. Existing voxel-based methods do not perform well enough to detect small objects that are difficult to detect such as pedestrians, motorcycles, and traffic cones. The main reasons are: (1) small objects such as pedestrians are smaller than vehicles, and LiDAR scans fewer valid points on the target, as shown in Figure 1; (2) the voxel-based method is characterized by the ability to encode the point cloud into voxels and average the corresponding features in the voxel grid. While this method provides superior computational efficiency, due to the small number of valid points for small objects, this method leads to a large loss of valid feature information for small objects, which affects the performance of small object detection. To address the challenge, we attempted to make it possible for the network to focus more on feature extraction from the foreground point cloud during voxelization. Attention mechanisms [10] have emerged as a particularly promising approach that capitalizes on their ability to enable models to focus on key information while avoiding the imposition of significant computational overhead. Specifically, the attention mechanism allows the neural network to concentrate on specific features of the input, and the effectiveness of attention remains consistent regardless of the input’s dimensionality. Hu et al. [11] introduced SENet, a network architecture derived from the SE block. The architecture specifically emphasizes channel dependencies and employs global average pooling to compute the weights assigned to each channel within the feature. Woo et al. [12] introduced the Convolutional Block Attention Module (CBAM), which builds upon the improvements of SENet. CBAM combines the individual advantages of spatial and channel attention mechanisms and further refines the feature map through the multiplication of the attention weights. Inspired by them, we propose a Dual-Attention Voxel Feature Extractor (DA-VFE), which combines pointwise attention and channelwise attention (Dual-Attention), and combines the Dual-Attention with the mean Voxel Feature Extractor (VFE) to extract features globally from the point cloud. The distribution of the points in the voxels is inhomogeneous, which is ameliorated by the DA-VFE, allowing for the refinement of more-representative point cloud information.

At the heart of the voxel-based method lies an interaction between two different backbones: a 3D Backbone and a 2D Backbone. The 3D Backbone is responsible for processing each voxel and extracting its unique features, which are then projected onto a Bird’s-Eye View (BEV) to generate a Two-Dimensional (2D) pseudo-image, which serves as the input to the 2D Backbone [13]. The 2D Backbone plays a crucial role in extracting the features from the pseudo-images, using a series of convolutions to generate qualitative suggestions and detection results. Typically, the 2D backbones used in voxel-based methods bear a striking resemblance to established models (e.g., VoxelNet [6] and SECOND [14]). However, it is worth noting the relative lack of focus on the 2D Backbone in most methods, which instead focus on optimizing the 3D Backbone to extract additional features from the underlying point cloud data. This focus has led researchers to overlook the potentially transformative impact of an enhanced 2D Backbone on the accuracy and reliability of object detection. Therefore, we designed a more-sophisticated 2D Backbone named the MFF Module, which consists of self-calibrated convolutions [15], coordinate attention [16], and a residual structure [17]. Our proposed 2D Backbone significantly expands the receptive field, enhances contextual information capture, and improves overall detection performance, particularly with regard to small objects. We evaluated our framework on the nuScenes [18] dataset. The experimental results showed that AMFF-Net achieved significant performance improvements compared to the baseline network CenterPoint [19] with an almost constant inference speed and a significant reduction in the overall number of parameters of the model, which demonstrated that our proposed framework can achieve performance improvements with reduced computational overhead.

Our main contributions can be summarized as follows:We designed a Dual-Attention Voxel Feature Extractor (DA-VFE) to integrate the Dual-Attention mechanism into the voxel-feature-extraction process, extract more-representative point cloud features, and reduce the information loss in the process of extracting the voxel features.We propose a novel 2D Backbone named the MFF Module, which extends the perceptual field of the 2D Backbone and can capture more contextual information to extract richer information with higher accuracy and robustness.The proposed network can achieve performance gains while reducing the computational overhead.

## 2. Related Work

In recent years, with the development of automated driving technology, the requirements for real-world 3D perception have been increasing. An emerging trend in 3D-object-detection research has seen a gradual shift away from RGB images, which can only provide limited 2D planar information, in favor of point cloud data, which offer a more-comprehensive and -accurate representation of the depth information. Typically, point-based methods leverage the features of the original point cloud data, with F-PointNet [20] representing a pioneering approach to applying PointNet [21], a 2D detection method predicated on cropped point clouds, to the realm of 3D object detection. The follow-up work [5] was a pioneering work that achieved good detection performance by the feature extraction of raw irregular point clouds. F-ConvNet [22] is optimized in more detail on F-PointNet. PointRCNN [23] segments the point cloud directly, generating a few qualitative three-dimensional candidate images directly in the point cloud, rather than projecting the point cloud into a pseudo-image. VoteNet [24] proposed the Hough voting strategy to optimize the point cloud data grouping. Furthermore, STD [25] introduced an innovative spherical anchor designed to generate highly accurate and reliable proposals. The 3D-SSD [26] proposed new sampling methods to better group object features. The IA-SSD [27] employs an instance-aware downsampling strategy to identify foreground points of interest. While this downsampling strategy effectively reduces redundant points, it also results in inevitable information loss. The point-based method retains the original geometric structure information, but suffers from the disadvantages of slow perception and high computation.

Voxel-based methods are dominant in the realm of 3D object detection. At the core of voxel-based methods is a complex voxelization process, which enables the input point cloud to be transformed into a voxel representation. This process is followed by the use of a 3D convolution to extract the voxel features across the scenario. The pioneering VoxelNet [6] for point cloud 3D object detection revolutionized the field by introducing a new method for dividing a point cloud into a large number of homogeneous voxels so that all points within a voxel can be converted into voxel features by a voxel-feature-coding layer. Then, the voxel features are used as the inputs to 3D Backbone. However, voxels are sparse in autonomous driving scenarios; there exist a large number of voxels that do not contain point clouds, and processing empty voxels can place a significant burden on the processor. In order to decrease the computational overhead, 3D sparse convolution is introduced by SECOND [14], which realizes efficient 3D convolution processing and greatly improves the inference speed of the network. PointPillars [7] represents a paradigm shift approach to voxel-based point cloud 3D object detection by further elongating voxels into pillars along the z-axis, dividing the point cloud into these pillars, and then, extracting features from the point cloud in these pillars to form a pseudo-BEV 2D image. This method avoids the use of low-efficiency 3D convolution, is faster in inference, and is popular in industry. Part-A2 [28] uses pointwise part location features as additional supervisory information, which is fused with pointwise semantic features to generate higher-quality 3D proposals. TANet [29] utilizes a sophisticated and innovative triple-attention mechanism to boost the robustness and accuracy of voxel feature learning. PV-RCNN [30] utilizes sparse convolution to extract high-level features from voxels to generate detailed suggestions. In this method, multiscale voxel features are encoded as keypoints, and the box is refined by a process of the aggregation of features from grid points located around the keypoints. The method preserves the structural information of the 3D point cloud by the additional branches of the key point cloud, which is a slower inference, even though it improves the detection performance. Voxel-RCNN [31] introduced a novel detector that utilizes only 3D voxel features for 3D object detection. This method utilizes voxel ROI pooling operations to further refine the anchor, and the inference speed of the method is dramatically improved compared to PV-RCNN [30]. Voxel ROI pooling does not require point information and avoids the interaction between the feature from point clouds and the feature from voxels. BtcDet [32] addresses the issue of shape incompleteness in point cloud data caused by occlusion and truncation by learning additional point cloud data that align with the target shape. It effectively complements the obscured shapes of the target. All the methods mentioned above use anchors to attach to objects. An anchor-free method was proposed for CenterPoint [19], which treats the target as a keypoint, predicts the center of the target, regresses to its orientation, and performs better in pedestrian detection compared to the anchor-based methods.

Through the analysis of voxel-based methods, the use of voxel feature encoding is a critical requirement in all methods for voxel-based 3D object detection. Therefore, we propose a Dual-Attention Voxel Feature Extractor (DA-VFE), which enables the learning of voxel-level feature representations that are both more robust and discriminative. In addition, this paper proposes a Multi-scale Feature Fusion Module (MMF Module), which contains self-calibrated convolution, coordinate attention, and a residual structure. Compared to the traditional RPN network, this Module boasts a larger receptive field, making it capable of capturing more contextual information. This, in turn, allows for the extraction of richer information and boosts detection performance for small objects. The proposed AMFF-Net is based on CenterPoint, and compared to CenterPoint, we boosted the detection performance of small objects and reduced the computational overhead while ensuring the overall accuracy and inference speed.

## 3. Method

Our model builds on the CenterPoint framework, using it as a baseline and modifying its architecture to create the proposed AMFF-Net. In this section, we provide a comprehensive overview of AMFF-Net, detailing its architecture and functionality, as shown in Figure 2. There are two main contributions: (1) we propose a Dual-Attention Voxel Feature Extractor (DA-VFE), which is embedded between voxelization and a 3D Backbone for enhancing voxel features and extracting more-representative original point cloud information; (2) we designed a novel 2D Backbone named the MFF Module, which mainly consists of self-calibrated convolutions, coordinate attention, and a residual structure, to improve the detection performance of small objects.

### 3.1. Voxelization

The voxelization is split into two phases: the first phase is called grouping, and the second phase is called sampling. The point cloud is initially divided into a certain number of voxels (dividing the space into a grid one by one, with a grid representing the point clouds in the grid), and these voxels are grouped; each point is assigned to a corresponding voxel based on its spatial coordinates. Then, random sampling of the points is performed. In addition to reducing the computational load, random sampling can effectively reduce the information bias caused by the uneven data of each voxel point cloud and improve the training efficiency. Then, the Voxel Feature Extractor (VFE) is utilized to extract the local features of every non-empty voxel, and each voxel is characterized by the average of the raw point cloud in every non-empty voxel; finally, a voxelwise feature is obtained.

### 3.2. Dual-Attention Voxel Feature Extractor

As shown in Figure 3, the Dual-Attention Voxel Feature Extractor (DA-VFE) integrates pointwise attention, channelwise attention, and VFE architectures with the aim of reducing the information loss during voxel feature extraction. Since the points are distributed in a disordered manner within each voxel, we designed a Dual-Attention mechanism combining pointwise and channelwise attention, which adaptively weighs the contributions of different perspectives.

#### 3.2.1. Pointwise Attention

For an arbitrary voxel grid $V^{t}\in R^{N\times C}$ in space, a max-pooling operation is first performed. Point features are aggregated by identifying and extracting the maximum value of the voxel $V^{t}$ across each channel via a max-pooling operation. As the first row of Figure 3 shows, the voxel $V^{t}$ is converted into a specific pointwise response $E^{t}\in R^{N\times 1}$. To more effectively leverage the shape features of the point cloud, we incorporated two fully connected layers, enabling it to learn a comprehensive and nuanced global pointwise response, denoted as
(1)$$S^{t}=W_{2}\sigma \left(W_{1}E^{t}\right)$$
where $W_{1}\in R^{r\times N}$ and $W_{2}\in R^{N\times r}$ denote the two essential weight parameters from the fully connected layer. σ denotes the ReLU activation function, and $S^{t}$ denotes the pointwise attention of $V^{t}$. The pointwise attention Module is designed to strengthen points that make a significant contribution to the overall analysis, while attenuating the impact of points that have a relatively small effect on the final result.

#### 3.2.2. Channelwise Attention

For each voxel grid $V^{t}\in R^{N\times C}$, in order to extract the channel features, the point features are aggregated through max-pooling across their respective channelwise dimensions. As shown in the second row of Figure 3, the maximum value of the voxel $V^{t}$ across every point dimension is obtained by the max-pooling operation, and the channel attention processed by the two fully connected layers is calculated as
(2)$$T^{t}=W_{2}^{{}^{\prime }}\sigma \left(W_{1}^{{}^{\prime }}{\left(U^{t}\right)}^{T}\right)$$
where the specific channel-level response $U^{t}\in R^{1\times C}$ is implemented by the max-pooling layer. The output of the channel-level attention $T^{t}\in R^{1\times C}$ is dependent on two fully connected layer parameters $W_{1}^{{}^{\prime }}\in R^{r\times C}$ and $W_{2}^{{}^{\prime }}\in R^{C\times r}$. The channelwise attention serves to highlight the significance of the feature channels present in every voxel.

The proposed Dual-Attention combines pointwise attention and channelwise attention. The attention matrix $M^{t}\in R^{N\times C}$ is obtained by multiplying the pointwise attention matrix $S^{t}$ and the channelwise attention. The output is
(3)$$M^{t}=\sigma \left(S^{t}\times T^{t}\right)$$
where σ denotes the sigmoid function, which maps real numbers to the interval [0,1].

The original voxel $V^{t}$ and the Dual-Attention $M^{t}$ are aggregated into a re-weighted feature $V_{s}^{t}\in R^{N\times C}$ to obtain the enhanced feature representation as
(4)$$V_{s}^{t}=M^{t}\cdot V^{t}$$
where · denotes elementwise multiplication. Then, all point cloud features within each voxel are aggregated using a max-pooling layer to obtain the feature representation of the voxel $F_{1}^{t}\in R^{1\times C}$.

Converting point features to voxel features with a mean Voxel Feature Extractor (VFE) is the most-useful method. For each non-empty voxel, the mean value is calculated for all the points $p_{i}$ inside, and the formula is
(5)$$F_{2}^{t}=\frac{\sum_{i=1}^{N^{t}}p_{i}^{t}}{N}$$
where the number $N^{t}$ denotes the number of points contained within the voxel and pi contains the four dimensional features xi,yi,zi,ri. As a result, the output of the Dual-Attention voxel feature extractor, denoted as $F^{t}\in R^{1\times C}$, can be produced by the connection of the meanvoxel feature $F_{2}^{t}$ and the Dual-Attention voxel feature $F_{1}^{t}$ as
(6)$$F^{t}=F_{1}^{t}+F_{2}^{t}$$
where + denotes the elementwise sum. The strengthened voxel features are used as the input for the 3D Backbone.

We introduced Dual-Attention (DA) into the Voxel Feature Extractor (VFE). The pointwise attention in DA aims to learn the point cloud comprehensively with global points accordingly, focusing on the points that contribute significantly to the overall analysis. In contrast, the channelwise attention in DA aggregates the features of the point cloud in their respective channel directions, focusing on the importance of the channel direction features in the point cloud within each voxel. By integrating DA into the VFE, we can add more feature information from the raw point cloud to the voxel-feature-extraction process, so that the VFE can pay more attention to the points in the point cloud data that have a high contribution to the detected objects (especially for small objects with only a few valid points) and improve the utilization of the information in the raw point cloud to reduce the loss of information in the voxel-feature-extraction process.

### 3.3. Multi-Scale Feature Fusion Module

After the 3D Backbone, a 3D feature map is obtained, and the feature map is compressed along the z-axis to obtain a pseudo-image of size 256×X/8×Y/8. The features of this pseudo-image in Bird’s-Eye View (BEV) are then extracted using the 2D Backbone. The original 2D Backbone extracts useful information on the BEV feature map by a series of 2D convolutions, and there exist two downsampling branch structures, corresponding to the existence of two deconvolution structures. The original Backbone structure is shown in Figure 4a. To be specific, in each downsampling branch, a 2× downsampling of the feature map is performed using one standard 3×3 convolution. Then, for the feature extraction, a sequence of five standard 3×3 convolutions is applied. The two obtained feature maps of different sizes are each upsampled to the same size using deconvolution and, then, concatenated in the channel dimension. Finally, the channel dimension is compressed by an additional standard 3×3 convolution.

Nonetheless, this feature-extraction approach suffers from a limited receptive field and cannot fully capture extensive contextual information, which is crucial for localizing small objects. Thus, we propose a novel 2D Backbone called the Multi-scale Feature Fusion (MFF) Module to improve the detection performance of small objects. As shown in Figure 4b, the MFF Module mainly consists of self-calibrating convolution, coordinate attention, and a residual structure. Specifically, we performed 1× and 2× downsampling using two normal 3×3 convolutions to obtain multi-scale feature maps. The coordinate attention mechanism is utilized after downsampling to accurately capture the object location during the channel-attention-construction process to extract more-informative feature representations. This attention mechanism produces more-representative feature representations, thereby increasing the recall of small objects. Then, the features are extracted using two consecutive self-calibrated convolutions, which build dependent calibration operations around each spatial location in remote spaces and between channels to establish large global receptive fields and enrich the contextual information of the features. Furthermore, we introduced a shortcut between the coordinate attention’s input and the second self-calibrating convolution’s output.

Self-calibrated convolution as an enhanced version of standard convolution was proposed by Liu et al. [15]. The receptive field of the convolutional layer is broadened by the internal communication of the self-calibrated convolution. Self-calibrated convolution builds remote spatial and channel-dependent calibration operations around each spatial location, enabling the extraction of more-robust feature representations without introducing additional parameters. Coordinate attention is a novel attention mechanism proposed by Hou et al. [16]. The coordinate attention mechanism splits the channel attention mechanism into two 1D features encoded for processing by aggregating two spatially oriented features. With this encoding method, it is possible to capture long-range dependencies along one spatial dimension while maintaining accurate positional information in another spatial dimension. This attention mechanism can embed coordinate information for input object features, which facilitates the localization of small objects. The architecture of the self-calibrated convolution and coordinate attention is shown in Figure 5a,b.

### 3.4. Loss Function

The AMFF-Net is based on CenterPoint for object detection. The loss function $L_{det}$ of the network consists of the center pointprediction loss $L_{k}$, the bias loss $L_{off}$ of the target center, and the loss $L_{size}$ of the target size. The focal loss function is used for center pointprediction [33,34].
(7)$$L_{k}=-\frac{1}{N}\sum_{xyc}^{}\left\{\begin{array}{cc} {\left(1-{\hat{Y}}_{xyc}\right)}^{\alpha }\log_{}{\hat{Y}}_{xyc} & \;\mathrm{if}\;Y_{xyc}=1\\ {\left(1-Y_{xyc}\right)}^{\beta }{\left({\hat{Y}}_{xyc}\right)}^{\alpha }\log_{}\left(1-{\hat{Y}}_{xyc}\right) & \;\mathrm{otherwise}\; \end{array}\right.$$
where ${\hat{Y}}_{xyc}$ is the predicted key point heat map value, Gaussian kernel $Y_{xyc}=exp\left(-\frac{{\left(x-{\tilde{p}}_{x}\right)}^{2}+{\left(y-{\tilde{p}}_{y}\right)}^{2}}{2\sigma_{p}^{2}}\right)$, and $\sigma_{p}$ is is an object-size-adaptive standard deviation [35]. A prediction ${\hat{Y}}_{xyc}=1$ corresponds to a detected keypoint, while ${\hat{Y}}_{xyc}=0$ is the background. α and β are hyperparameters of the focal loss, and N is the number of positive samples in the ground truth sample, which is used to normalize all positive focal loss to 1.

Each key point $\tilde{p}$ needs to predict a center position bias $\hat{O}$. The bias loss of the target center is calculated using the L1 loss function, defined as
(8)$$L_{off}=\frac{1}{N}\sum_{p}^{}\left|{\hat{O}}_{\tilde{p}}-\left(\frac{p}{R}-\tilde{p}\right)\right|$$
where ${\hat{O}}_{\tilde{p}}\in R^{\frac{W}{R}\times \frac{H}{R}\times 2}$ denotes the predicted bias of the keypoint coordinates; *p* is the ground truth keypoint coordinates; *R* is the downsampling factor; $\;\tilde{p}$ is the predicted keypoint coordinates.

In addition, for each key point, its width and height dimensions ${\hat{S}}_{pk}\in R^{\frac{W}{R}\times \frac{H}{R}\times 2}$ need to be predicted; its center point lies at $p_{k}=\left(\frac{x_{1}^{\left(k\right)}+x_{2}^{\left(k\right)}}{2},\frac{y_{1}^{\left(k\right)}+y_{2}^{\left(k\right)}}{2}\right)$; the target size loss is calculated using the same L1 loss function, defined as
(9)$$L_{size}=\frac{1}{N}\sum_{k=1}^{N}\left|{\hat{S}}_{pk}-S_{k}\right|$$
where $S_{k}=\left(x_{2}^{\left(k\right)}-x_{1}^{\left(k\right)},y_{2}^{\left(k\right)}-y_{1}^{\left(k\right)}\right)$ is the width and height dimension of the ground truth keypoint.

The overall loss function $L_{det}$ is the sum of the predicted loss at the center point, the bias loss at the target center, and the loss at the target size, each with a corresponding weight.
(10)$$L_{det}=L_{k}+\lambda_{off}L_{off}+\lambda_{size}L_{size}$$
where $\lambda_{off}$ and $\lambda_{size}$ are the weights of the bias loss of the target center and the loss of the target size, respectively.

## 4. Experiments

### 4.1. Experiment Details

#### 4.1.1. Dataset

**nuScenes dataset:** The nuScenes dataset [18] contains 1000 driving scenarios, with 700, 150, and 150 scenarios for training, validation, and testing, respectively. The nuScenes uses a 32-lane rotating LiDAR with a 20 Hz sampling frequency, which can generate almost 30,000 points per frame. The dataset is a box annotated every 10 frames to provide calibrated vehicle posture information for each LiDAR frame. The annotations include 10 object categories with a total of 40,000 annotated frames. The official evaluation metric is the average accuracy in each category. For 3D object detection, the main evaluation metrics are the mean Average Precision (mAP) and the nuScenes Detection Score (NDS). The mAP uses < 0.5 m, 1 m, 2 m, and 4 m bird’s-eye view center distances instead of the standard box overlap. The NDS is a weighted average of the mAP and other attribute quantities, including translation, scale, direction, velocity, and other box attributes [18].

**KITTI dataset:** The KITTI dataset [36] provides 7481 training samples and 7518 test samples. In the experiments, the KITTI training set was divided into a training set and a validation set in a ratio of 0.5:0.5. For the experimental studies on the validation set, we used the training set for training. The KITTI dataset provides three categories of recognized object labels, which are cars, pedestrians, and cyclists. The IoU thresholds for these three categories of objects were set as follows: 0.7 for cars and 0.5 for pedestrians and cyclists. Each category involves three difficulty levels (Easy, Moderate, and Hard) depending on the degree of occlusion and the percentage of truncation of the object. The Average Precision (AP) over 40 recall positions and the mean AP (mAP) of three difficulty levels were used as our evaluation metrics.

#### 4.1.2. Data Augmentation

Throughout the training phase, we leveraged a range of commonly used data augmentation means that are popular in the field of 3D object detection: random flips along the x- and y-axes with global scaling using a random factor from [0.95, 1.05] and random rotations along the z-axis ranging from [−π/8, π/8] [37]. We also used ground truth sampling on nuScenes to handle the long-tail class distribution [14], which copies and pastes the points in the annotation box from one frame to another. In this way, randomly categorized ground truth objects are selected and replicated from a variety of different scenarios and, then, placed in the current training scenario.

#### 4.1.3. Parameter Settings

For the nuScenes dataset experiments, we set the detection range to [−54 m, 54 m] for the x- and y-axes and [−5 m, 3 m] for the z-axis. The voxel size of the model was (0.075 m, 0.075 m, 0.2 m), so the whole scene was divided into 1440 × 1440 × 40 voxels. Since the number of voxels in the whole voxel space was too large, we limited the number of voxels to the range [120,000, 160,000], and a limit of 10 LiDAR points per voxel was enforced to limit the maximum number of points contained in a single voxel, in order to reduce the hardware resource consumption and improve the training speed.

For the KITTI dataset experiments, we set the detection range to [0 m, 70.4 m] for the x-axis, [−40 m, 40 m] for the y-axis, and [−3 m, 1 m] for the z-axis. The voxel size of the model was (0.05 m, 0.05 m, 0.1 m), so the whole scene was divided into 1600 × 1408 × 40 voxels. The maximum number of non-empty voxels was set to 40,000, and each voxel was sampled at a maximum of 5 points.

#### 4.1.4. Training Details

AMFF-Net was trained in an end-to-end way. The AdamW [38] optimizer was used to optimize the model, and the learning rate in the training stage was adjusted using a single-cycle strategy [39] with a maximum learning rate of 0.003, a weight decay of 0.01, a division factor of 10, and a momentum range of [0.85, 0.95]. We trained the model on 2 RTX 4090 GPUs, and the complete training process took about 48 h. To prevent overfitting, we searched for the optimal tuning parameters based on the validation set during the training process, aiming to keep the model in its best state. A common practice [18,26,37,40] in nuScenes is to transform and merge Lidar points from unannotated frames into the annotated frames that follow them. This produces a denser point cloud, and this practice was followed during both the training and experiments.

### 4.2. Experimental Results

**Results on nuScenes dataset:** Table 1 shows the performance of the other algorithms and the proposed AMFF-Net on the nuScenes dataset. The proposed AMFF-Net was compared with other algorithms for detection in each of the 10 categories. Specifically, in most categories, the proposed AMFF-Net achieved superior performance, including car, truck, trailer, bus, construction vehicle, pedestrian, motor, bicycle, and traffic cone. Especially for motor, bicycle, traffic cone, and pedestrian, these small object categories showed a substantial improvement in detection accuracy. It is worth mentioning that the AMFF-Net performed even better than the multimodal PointPainting [40] for detection in each category. The effectiveness of the model was verified by comparing it with other methods.

Our baseline network was CenterPoint, and our model was compared with the baseline in terms of the error rate metrics [18], inference speed, and model parameters. The Average Translation Error (ATE) is the two-dimensional Euclidean center distance (in meters). The Average Scale Error (ASE) is 1-IoU, where the IoU is the 3D intersection ratio after angular alignment. The Average Orientation Error (AOE) is the minimum yaw angle difference between the predicted and true values. The Average Velocity Error (AVE) is the L2 parametrization of the 2D velocity difference (m/s). The Average Attribute Error (AAE) is defined as 1-acc, where acc is the category classification accuracy. As shown in Table 2, our AMFF-Net outperformed the baseline on most metrics, and the inference speed was basically the same compared to the baseline; but, the overall number of model parameters was significantly reduced, which indicated that our proposed model had reduced computational overhead compared to the baseline.

**Results on KITTI dataset:** The proposed method was tested on the validation set for comparison and analysis. The performance on the validation set was calculated with the AP setting of 40 recall positions. We referenced the R40 APs of VoxelNet, TANet, PV-RCNN, BtcDet [32], and MA-MFFC [47] in their papers and the R40 APs of SECOND, PointRCNN, Part-A2, PointPillars, and Voxel R-CNN in the results of the officially released code. The results are shown in Table 3. It can be seen that the 3D mean average precision of AMFF-Net was 0.87% higher than the method MA-MFFC. Although our detection accuracy at the easy level of cyclists was lower than the method MA-MFFC, our method outperformed it at both the moderate and hard levels. Moreover, the detection accuracy was better than the method MA-MFFC at all three detection difficulty levels of cars and pedestrians.

### 4.3. Ablation Studies

This section outlines the impact of various components of AMFF-Net on the detection performance of this network, including the effect on different components within the DA-VFE and MFF Module. We conducted an extensive series of ablation studies on the validation set of the nuScenes dataset. The results were evaluated using the metrics AP, mAP, and NDS for 3D detection with the nuScenes dataset.

Table 4 shows the effect of the DA-VFE and MFF Module on the detection performance of the proposed AMFF-Net. The baseline was the CenterPoint network, and the mAP and NDS increased to 59.7% and 67.0%, respectively, with the addition of the DA-VFE. After replacing the baseline 2D Backbone with the MFF Module, the mAP improved to 60.1% and the NDS improved to 66.9%. Combining these two components, finally, our AMFF-Net achieved a mAP of 60.4%, as well as an NDS of 67.3%. Among them, the detection of small objects such as pedestrian, motorcycle, bicycle, and traffic cone improved significantly, by 1.4%, 2.0%, 5.6%, and 4.3%, respectively.

#### 4.3.1. DA-VFE

The baseline was the CenterPoint network. The Voxel Feature Extractor (VFE) of the original network was replaced with the Dual-Attention Voxel Feature Extractor (DA-VFE). When only pointwise attention was combined with the VFE, the mAP and NDS improved to 58.7% and 66.5%, respectively. This demonstrated that the pointwise attention was capable of enhancing the points with a high contribution while attenuating points with a lower contribution. When only channelwise attention was combined with the VFE, the mAP and NDS improved to 58.8% and 66.4%, respectively. The channelwise attention better represented the importance of the feature channels in each voxel. We refer to the parallel attention fusion of pointwise attention and channelwise attention as Dual-Attention (DA). When they were combined, DA produced a mAP and NDS of 59.7% and 67.0%, respectively. For the small object categories of pedestrians, motorcycles, bicycles, and traffic cones, the AP improved by 0.9%, 1.4%, 2.7%, and 2.6%, respectively, compared to the baseline. This indicated that DA was very important for the rational utilization of both spatial and channel information, which can highlight the small object feature information with a small number of valid point clouds and improve the performance of small object detection, thereby improving the robustness of 3D object detection. Table 5 shows the results.

#### 4.3.2. MFF Module

The baseline was the CenterPoint network. We modified the 2D Backbone of the original network by adding self-calibrated convolutions, a residual structure, and a coordinate attention mechanism to it. As shown in Table 6, the use of self-calibrated convolution enables the extraction of features across different scales; the mAP improved to 59.0% compared to the baseline. More contextual information can be extracted using self-calibrating convolution because of its property of extending the receptive field. Through adding the coordinate attention to capture the exact position of the object during the construction of the channel attention, the mAP and NDS were improved to 59.7% and 66.9%, respectively. Incorporating a residual structure led to an increase in the mAP to 60.1% and the NDS to 66.9%. The final detection performance of using the MFF Module as the 2D Backbone to extract features was improved in the small object categories compared to the baseline; especially, for bicycle and traffic cone, the improvement was huge: the mAP improved by 4.3% and 3.2%, respectively. This showed that our MFF Module can significantly boost the accuracy of small object detection while ensuring the overall accuracy improvement.

## 5. Conclusions

In this paper, we introduced an advanced 3D object detector named AMFF-Net, which was specifically designed to detect and analyze 3D objects from the original point cloud. We designed a Dual-Attention Voxel Feature Extractor (DA-VFE) to integrate the Dual-Attention mechanism into the voxel-feature-extraction process to extract more-useful information from the original point cloud so that the voxel feature extractor can pay more attention to the points in the point cloud data that have a high contribution to the detected objects and obtain more-robust voxel features. The DA-VFE can greatly increase a model’s ability to detect a target from a small number of points, which is especially critical for small objects with only a few valid points. We designed a Multi-scale Feature Fusion Module (MFF Module) consisting of a residual structure, self-calibrated convolutions, and a coordinate attention mechanism as the 2D Backbone of the network to aggregate richer feature information and improve the detection performance. The MFF Module extends the receptive field of the model to capture a wide range of contextual information, which, together with the coordinate feature information, is an advantageous factor for localizing and detecting small objects. The experiments on the nuScenes dataset showed that the proposed AMFF-Net significantly boosted the detection performance of small objects while reducing the computational overhead, achieving a good balance between performance and inference speed. AMFF-Net also achieved outstanding performance on the KITTI dataset. Limited by the performance of the hardware, our method is currently not completely applicable to real-time driving scenarios. In future work, we will continue to improve this algorithm and optimize the network architecture to further reduce the network inference time, aiming to improve the usability of the model in real-time driving scenarios while maintaining high detection accuracy.

## Figures and Tables

**Figure 1 sensors-23-09319-f001:**
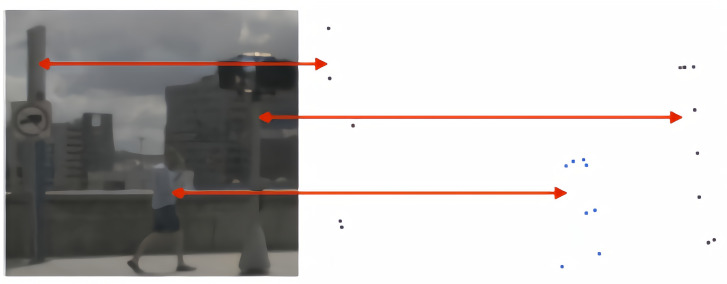
An autonomous-driving-detection scenario. LiDAR scans few valid points for small objects such as pedestrians or utility poles.

**Figure 2 sensors-23-09319-f002:**
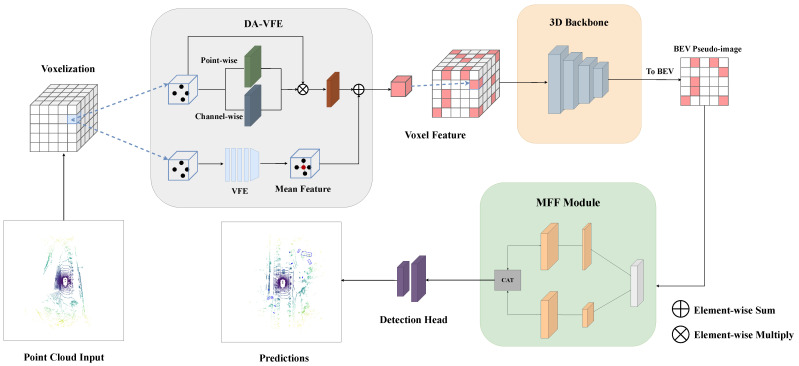
Architecture of the proposed AMFF-Net.

**Figure 3 sensors-23-09319-f003:**
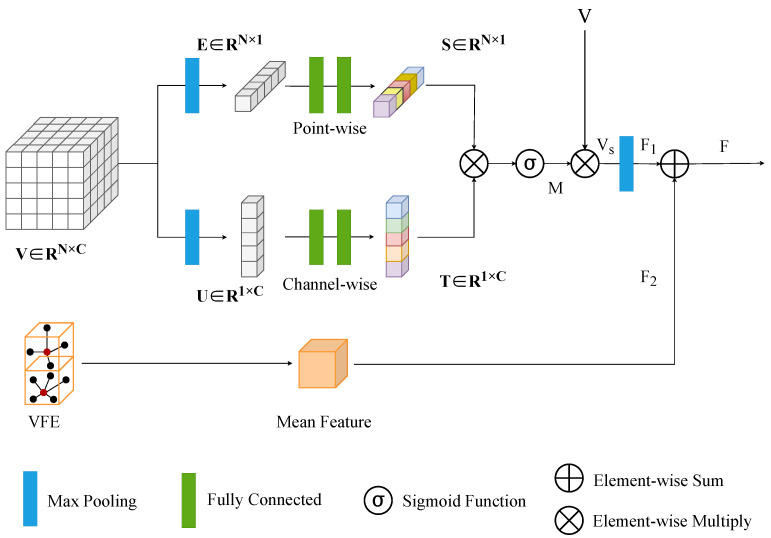
Architecture of the DA-VFE. The two lines in the top column represent the pointwise and channelwise attention, respectively, which form the Dual-Attention. The bottom of the figure indicates the mean extraction of voxel features.

**Figure 4 sensors-23-09319-f004:**
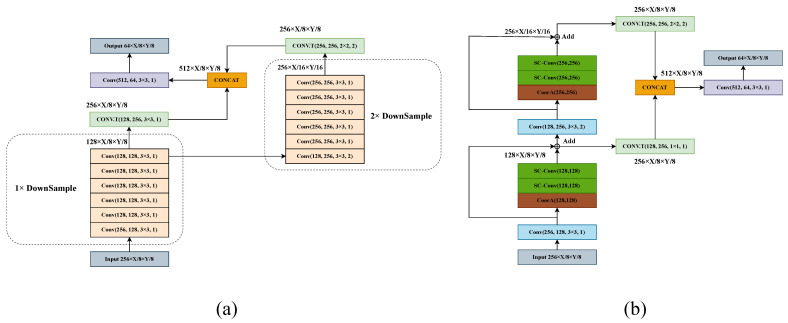
(**a**) Architecture of original 2D Backbone. (**b**) Architecture of MFF Module.

**Figure 5 sensors-23-09319-f005:**
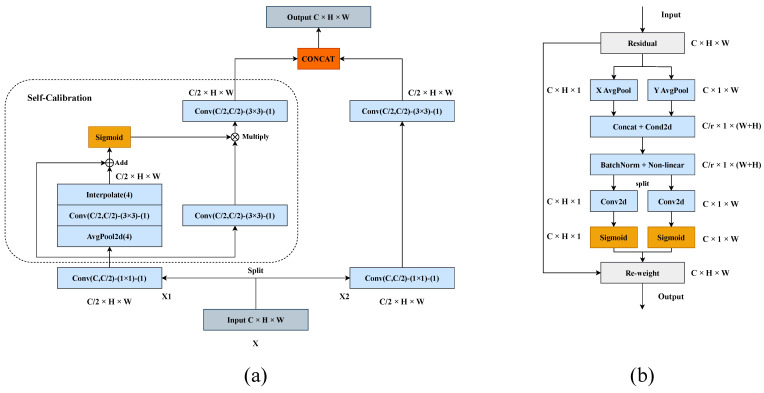
(**a**) Architecture of Self−Calibrated Convolution (SC−Conv). (**b**) Architecture of coordinate attention.

**Table 1 sensors-23-09319-t001:** Performance comparison of 3D detection on nuScenes test set. Results in bold denote the best of all of the methods. ${}^{‡}$ means flipping and rotation testing time augmentations. Abbreviations: Construction Vehicle (CV), Pedestrian (Ped), Motorcycle (Motor), and Traffic Cone (TC).

Methods	mAP	NDS	Car	Truck	Bus	Trailer	CV	Ped	Motor	Bicycle	TC	Barrier
PointPillars [7]	30.5	45.3	68.4	23.0	28.2	23.4	4.1	59.7	27.4	1.1	30.8	38.9
WYSIWYG [41]	35.0	41.9	79.1	30.4	46.6	40.1	7.1	65.0	18.2	0.1	28.8	34.7
GVNet [42]	35.4	/	76.2	29.7	42.3	22.1	/	59.2	20.7	0.8	/	32.0
SMS-Net [43]	43.7	57.6	80.3	49.5	61.3	35.1	12.1	71.2	28.9	4.6	46.0	34.7
PMPNet [44]	45.4	53.1	79.7	36.6	47.1	43.1	18.1	76.5	40.7	7.9	58.8	48.8
PointPainting [40]	46.4	58.1	77.9	35.8	36.2	37.3	15.8	73.3	41.5	24.1	62.4	60.2
SSN [45]	46.4	58.1	80.7	37.5	39.9	43.9	14.6	72.3	43.7	20.1	54.2	56.3
MFFFNet-CP [46]	46.6	/	79.7	46.8	64.9	32.5	8.9	74.1	36.5	14.1	51.6	56.5
CBGS [37]	52.8	63.3	81.1	48.5	54.9	42.9	10.5	80.1	51.5	22.3	70.9	65.7
CenterPoint [19]	58.0	65.5	84.6	51.0	60.2	53.2	17.5	83.4	53.7	28.7	76.7	70.9
CenterPoint ${}^{‡}$ [19]	60.3	67.3	85.3	53.5	63.6	56.1	20.0	84.6	59.4	30.7	78.4	**71.1**
AMFF-Net (Ours)	60.8	67.7	85.9	53.7	64.9	56.2	18.7	85.8	60.4	32.1	79.9	70.1
AMFF-Net (Ours) ${}^{‡}$	**62.8**	**69.2**	**86.5**	**55.9**	**66.8**	**58.5**	**21.3**	**86.6**	**63.6**	**35.6**	**82.0**	71.0

**Table 2 sensors-23-09319-t002:** Comparison with the baseline for 3D detection on the nuScenes validation set for different performance metrics. ${}^{‡}$ means flipping and rotation testing time augmentations. ↑ is for higher is better, and ↓ is for lower is better.

Methods	mAP↑	NDS↑	mATE↓	mASE↓	mAOE↓	mAVE↓	mAAE↓	FPS↑	Parameters↓
CenterPoint (Baseline)	58.3	66.1	29.4	25.7	30.2	26.4	19.1	13.1	8.94 M
AMFF-Net (Ours)	60.4	67.3	29.3	25.5	30.1	25.1	18.7	13.3	6.91 M
CenterPoint (Baseline) ${}^{‡}$	60.7	67.9	27.8	25.0	28.2	24.4	19.2	4.2	8.94 M
AMFF-Net (Ours) ${}^{‡}$	62.5	68.9	27.7	25.0	29.4	23.0	18.8	4.2	6.91 M

**Table 3 sensors-23-09319-t003:** Performance comparison of 3D detection. Average Precision (AP) and mean Average Precision (mAP) for 3D bounding boxes on the KITTI validation set. Results in bold denote the best of all of the methods.

Methods	Cars			Cyclists			Pedestrians			3D mAP
	**Easy**	**Moderate**	**Hard**	**Easy**	**Moderate**	**Hard**	**Easy**	**Moderate**	**Hard**	
VoxelNet	81.97	65.46	62.85	67.17	47.65	45.11	57.86	53.42	48.87	58.93
PointPillars	87.50	77.01	74.77	83.65	63.40	59.71	66.73	61.06	56.50	70.03
SECOND	89.05	79.94	77.09	82.96	61.43	59.15	55.94	51.14	46.17	66.99
PointRCNN	88.26	77.73	76.67	82.76	62.83	59.62	65.62	58.57	51.48	69.28
Part-A2	89.47	79.47	78.54	88.31	70.14	66.93	66.89	59.68	54.62	72.67
PV-RCNN	92.10	84.36	82.48	88.88	71.95	66.78	64.26	56.67	51.91	73.26
TANet	88.21	77.85	75.62	85.98	64.95	60.40	70.80	63.45	58.22	71.72
Voxel R-CNN	92.64	85.10	82.84	92.93	75.03	70.81	69.21	61.98	56.33	76.31
BtcDet	**93.15**	**86.28**	83.86	91.45	74.70	70.08	69.39	61.19	55.86	76.21
MA-MFFC	92.60	84.98	83.21	**94.78**	75.72	72.28	71.33	63.84	58.63	77.48
AMFF-Net (Ours)	92.67	85.79	**84.17**	92.56	**76.97**	**72.74**	**73.80**	**64.93**	**61.52**	**78.35**

**Table 4 sensors-23-09319-t004:** Ablation experiments of the proposed method with different components on nuScenes validation set.

Methods	Ped	Motor	Bicycle	TC	mAP	NDS
Baseline	84.4	58.7	39.7	68.4	58.3	66.1
+DA-VFE	85.3	60.1	42.4	71.0	59.7	67.0
+MFF	85.7	59.7	44.0	71.6	60.1	66.9
Ours	85.8	60.7	45.3	72.7	60.4	67.3

**Table 5 sensors-23-09319-t005:** Ablation experiments on the effects of pointwise attention, channelwise attention, and the proposed Dual-Attention on nuScenes validation set.

Methods	Ped	Motor	Bicycle	TC	mAP	NDS
Baseline	84.4	58.7	39.7	68.4	58.3	66.1
PA	85.3	59.8	40.8	68.9	58.7	66.5
CA	85.0	57.8	40.5	70.6	58.8	66.4
DA	85.3	60.1	42.4	71.0	59.7	67.0

**Table 6 sensors-23-09319-t006:** Ablation experiments on the effects of MFF Module on nuScenes validation set. SC-Conv denotes the Self-Calibrated Convolution. CA denotes the Coordinate Attention. Res denotes the Residual structure.

Baseline	SC-Conv	CA	Res	Ped	Motor	Bicycle	TC	mAP	NDS
✓				84.4	58.7	39.7	68.4	58.3	66.1
✓	✓			85.4	57.9	40.9	70.1	59.0	66.3
✓	✓	✓		85.7	59.7	40.4	70.6	59.7	66.9
✓	✓	✓	✓	85.7	59.7	44.0	71.6	60.1	66.9

## Data Availability

Publicly available datasets were analyzed in this study. These data can be found here: https://www.nuscenes.org/nuscenes (accessed on 20 October 2023) and https://www.cvlibs.net/datasets/kitti (accessed on 20 October 2023).

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
