# Peer review of "AMFF-Net: An Effective 3D Object Detector Based on Attention and Multi-Scale Feature Fusion"

_sensors, 2023, doi:10.3390/s23239319_

Round 1
Reviewer 1 Report
Comments and Suggestions for Authors
A well-done work that provides new insights into LiDAR-related practices. Multi-scale feature fusion network with attention mechanisms is an emerging approach of significant interest, and I appreciate the authors' work bringing about applications related to LiDAR sensing. I have two comments and suggestions as follows:
(1) The network architecture is relatively complex. Can the proposed method apply to real-time use cases?
(2) Have the authors considered the risk of overfiting? The training data sets are limited as described in the paper, and there is considerable similarity between the test and training sets. Please add some clarity regarding this issue.
Reviewer 2 Report
Comments and Suggestions for Authors
In this paper, the authors propose an Attention-based and Multiscale Feature Fusion Network (AMFF-Net), which utilizes a Dual-Attention Voxel Feature Extractor (DA-VFE) and a Multi-scale Feature Fusion (MFF) Module to improve the precision and efficiency of 3D object detection.
-
Please explain more about the proposed DA-VFE module, including how it could reduce information loss.
-
I found some typos existing in the manuscript. The English needs further improvement. For example, ground true (It should be ground truth).
-
Please explain the regression and jittering loss functions in more detail.
-
I find some typos existing in the manuscript. English needs further improvement. I encourage the authors to have their manuscript proof-edited to enhance paper presentation levels.
Reviewer 3 Report
Comments and Suggestions for Authors
The article focuses on detecting 3D objects using a mix of techniques: Attention and Multi-Scale Feature. The English is good. The structure is good. The problems:
1. I see a similar article on with the exact focus which was already published in Sensors:
Liu, M., Ma, J., Zheng, Q., Liu, Y., & Shi, G. (2022). 3D Object Detection Based on Attention and Multi-Scale Feature Fusion. Sensors, 22(10), 3935. Could you please write a detailed difference between these 2, and add it to your state of the art (SOA)? 2. The SOA is weak. 30 something references are not enough. Add more references, but be careful not to cite several at a time. Cite each and talk about the added value related to your study. 3. The Future work is non-existent, and Conclusions are short, compared to the size on the article. 4. Using only the nuScenes dataset seems insufficient. Also try to include the KITTI dataset used by the other guys and compare your work with theirs. Comments on the Quality of English LanguageThe English is decent.
Round 2
Reviewer 3 Report
Comments and Suggestions for Authors
Thank you for improving the paper. It seems ready to be published.